# Correcting Biased Centered Kernel Alignment Measures in Biological and Artificial Neural Networks

**Alex Murphy**
Department of Computing Science
Department of Psychology
University of Alberta
Alberta Machine Intelligence Institute
amurphy3@ualberta.ca

**Joel Zylberberg**
Department of Physics & Astronomy
University of York
joelzy@yorku.ca

**Alona Fyshe**
Department of Computing Science
Department of Psychology
University of Alberta
Alberta Machine Intelligence Institute
alona@ualberta.ca

## Abstract

Centred Kernel Alignment (CKA) has recently emerged as a popular metric to compare activations from biological and artificial neural networks (ANNs) in order to quantify the *alignment* between internal representations derived from stimuli sets (e.g. images, text, video) that are presented to both systems (Sucholutsky et al., 2023; Han et al., 2023). In this paper we highlight issues that the community should take into account if using CKA as an alignment metric with neural data. Neural data are in the *low-data high-dimensionality* domain, which is one of the cases where (biased) CKA results in high similarity scores even for pairs of random matrices. Using fMRI and MEG data from the THINGS project (Hebart et al., 2023), we show that if biased CKA is applied to representations of different sizes in the *low-data high-dimensionality* domain, they are not directly comparable due to biased CKA's sensitivity to differing feature-sample ratios and not stimuli-driven responses. This situation can arise both when comparing a pre-selected area of interest (e.g. ROI) to multiple ANN layers, as well as when determining to which ANN layer multiple regions of interest (ROIs) / sensor groups of different dimensionality are most similar. We show that biased CKA can be artificially driven to its maximum value when using independent random data of different sample-feature ratios. We further show that shuffling sample-feature pairs of real neural data does not drastically alter biased CKA similarity in comparison to unshuffled data, indicating an undesirable lack of sensitivity to stimuli-driven neural responses. Positive alignment of true stimuli-driven responses is only achieved by using *debiased* CKA. Lastly, we report findings that suggest biased CKA is sensitive to the inherent structure of neural data, only differing from shuffled data when debiased CKA detects stimuli-driven alignment.

## 1 Introduction

The goal of measuring (or inducing) alignment between biological and artificial neural networks relies on the selection of a metric that can correctly quantify representational alignment between two systems (Sucholutsky et al., 2023), while also being invariant to transformations that can cause similar representations to appear unrelated. CKA has recently gained popularity as a metric to study alignment between brain responses and deep learning models (Sucholutsky et al., 2023; Han et al., 2023). Furthermore, recent work to create more brain-like ANNs has used CKA in a loss function to

force models to exhibit more brain-like responses that maximize CKA-alignment to held-out neural data, resulting in models that are more robust to adversarial attacks (Dapello et al., 2023).

In the context of modern deep learning, it is common to have dataset sizes multiple orders of magnitude higher than what can be expected for neural data (Smucny et al., 2022). Therefore, when studying the alignment between ANNs and brains, similarity measures are applied to matrices which contain many more columns (features) than rows (samples). Previous research has noted that this situation is where standard ("biased") CKA becomes an unreliable alignment metric that can potentially lead to erroneous conclusions (Kleemann et al., 2007; Smilde et al., 2008). This work demonstrates this unreliability in a practical set of experiments, and promotes the adoption of a debiasing step put forward in Kornblith et al. (2019), which we show is robust to the aforementioned weakness. Though Kornblith et al. (2019) published their work in 2019, biased CKA has still appeared in publications. We further highlight the importance of using both *random* and *shuffled* controls when quantifying alignment between neural networks and brains.

We first examine the sensitivity of biased CKA to alignment of two random matrices, where one is of a fixed shape and we systematically increase the feature dimension of the other. We compare three alignment metrics: (i) biased CKA, (ii) debiased CKA and (iii) the RV2 Coefficient. We then examine how varying feature dimensionality alone can result in potentially misleading conclusions by contrasting biased CKA and debiased CKA across two CNN models (ResNet18 & CORnet-S), two brain data modalities (fMRI & MEG) over the same image stimuli set. We compare across three experimental manipulations: (i) using the correct data order (where samples and features are matched correctly), (ii) shuffled neural data and (iii) random data. The final section outlines interesting findings regarding biased CKA's insensitivity to shuffled data, in which we show that only in the debiased version do we see sensitivity to true sample-feature pairs. Additionally, we suggest that biased CKA is sensitive to the inherent structure of neural data responses and can thus be decomposed into such a component, in addition to true stimuli-driven responses detectable using debiased CKA.

The primary contributions of this work are:

1. We provide a systematic investigation of CKA for the alignment of biological and artificial neural networks, arguing for the importance of using random and shuffled controls and of applying the debiasing step to avoid erroneous conclusions.

2. We show that without debiasing, CKA alignment of data matrices that have a larger number of columns than rows is an unreliable quantification of similarity. This situation is a common occurrence when comparing ANN models with neural data when the number of parameters between ANN layers or neural data set sizes are not equal.

3. We show how biased CKA can give the incorrect impression that fMRI / MEG have similar alignment scores across multiple ANN layers, which the debiased version of CKA successfully detects as erroneous.

4. Our experiments support the idea that biased CKA is decomposable into at least two components: one that is sensitive to the generalized structure of neural data (e.g. prototypical fMRI responses to image viewing) and one that is sensitive to true stimuli-driven responses, while debiased CKA is only sensitive to the latter component.

## 2 CKA OVERVIEW

### 2.1 MATHEMATICAL FORMULATION & PROPERTIES

Centered Kernel Alignment (Cortes et al., 2012; Kornblith et al., 2019) is a metric used to quantify the similarity of representational structure between two matrices, each of which must be matched for the number of samples but can differ in the dimensionality of each feature space, i.e. $X \in \mathbb{R}^{N \times P1}$, $Y \in \mathbb{R}^{N \times P2}$, but $P1 \neq P2$. Unlike correlation/cosine-based measures of matrix similarity (e.g. Representational Similarity Analysis (Kriegeskorte et al., 2008)), CKA quantifies similarity using the dot-product operation. A generalization of dot-product similarity for centered matrices is the Hilbert-Schmidt Independence Criterion (HSIC), which is a kernel-based operation. Namely, if $K$ and $L$ represent the output of two kernel operations on the original data, the empirical estimator for HSIC is given by the following, where $H$ is a centering matrix:

$$\text{HSIC}(K, L) = \frac{1}{(n-1)^2} \text{tr}(KHLH) \tag{1}$$

CKA is the extension of HSIC through the application of a normalization step in order to introduce invariance to isotropic scaling:

$$\text{CKA}(K, L) = \frac{\text{HSIC}(K, L)}{\sqrt{\text{HSIC}(K, K)\text{HSIC}(L, L)}} \tag{2}$$

Regarding the selection of an appropriate kernel in CKA, it is virtually ubiquitous to use CKA with a linear kernel since initial experiments demonstrated that it performed comparatively well to non-linear kernels (Han et al., 2023; Davari et al., 2022a). We only consider the linear variant in this work.

The invariance properties of CKA are important to consider with regard to the other similarity metrics to which it has previously been compared. CKA is invariant to orthogonal transformations and isotropic scaling, but not to invertible linear transformations, unlike other methods such as Canonical Correlation Analysis (CCA). Kornblith et al. (2019) showed that several common similarity metrics were unable to detect similarity between corresponding layers of neural networks trained using the same architecture and hyperparameters, but different random seeds. In contrast, CKA was able to detect this similarity. Operations such as batch normalization apply an invertible linear transformation that changes ANN behavior, and it's not clear that a similarity metric should be invariant to such operations (Thompson et al., 2019). Other interesting findings using CKA to provide insights into the representational structure in ANNs were that wide networks learn similar representations and that saturation of feature channels is more commonly a feature of higher layers rather than lower layers (Nguyen et al., 2021). We next examine some of the criticisms of CKA and the attempts to address identified weaknesses.

## 2.2 CRITICISMS

Davari et al. (2022a;b; 2023) show that CKA is sensitive to subset translations, and that targeted attacks on the metric can artificially and arbitrarily modulate CKA similarity values, without changing the functional behaviour of a neural network. Thompson et al. (2019) point out that the standard implementation of CKA described above is highly sensitive to dataset size discrepancies, i.e. as the feature-sample ratio increases, CKA will tend towards $1.0$ even for random, unrelated matrices. This is a situation that is common when dealing with neural data, which often results in large feature dimensionalities (e.g. fMRI) from costly experiments, which restricts the amount of data that can be collected. This results in a comparatively lower number of samples that can be concurrently presented to both biological and artificial neural networks. Yao et al. (2020) reported that this issue was also present in biological analysis, e.g. transcriptomics studies, where feature dimensionality of gene expression profiles is much larger than the number of collected samples. Linear CKA is equivalent to a metric introduced in Robert & Escoufier (2018) called the RV Coefficient (Thompson et al., 2019; Klabunde et al., 2023), which Smilde et al. (2008) reformulated in terms of random matrices and showed that this metric can be approximated by the following formula, where $N$ denotes the number of samples and $P_1$, $P_2$ denote the feature dimensionality of each matrix:

$$\text{RV}(A, B) \approx \frac{P_1 P_2}{\sqrt{(P_1^2 + (N+1)P_1)(P_2^2 + (N+1)P_2)}} \tag{3}$$

By subtracting the diagonal elements off the Gram matrix, namely $AA^T - diag(AA^T)$, we eliminate the dependence of the metric value on the dimensionalites of $P_1$ and $P_2$, which are contained within the diagonal of the matrix. This results in the modified RV Coefficient (RV2) that is no longer sensitive to sample-feature ratio discrepancies.

In an updated version of their paper, Kornblith et al. (2019) point out that earlier work (Song et al., 2007) derived an unbiased estimator of HSIC by recognizing that HSIC can be formulated as a U-Statistic. They substituted a more numerically stable formation of this estimator (See A.5)) from Székely & Rizzo (2014) into the kernel-centering step of CKA, resulting in a *debiased* version,

which fixed the aforementioned issues. The range of the debiased version of CKA is $[-1, 1]$, unlike biased CKA which is in $[0, 1]$.

## 2.3 Applications to Neuroscience

Linear CKA is similar to Representational Similarity Analysis (RSA), a common tool in studying neural representations between different biological / artificial systems (Kriegeskorte et al., 2008). Whereas CKA implements dot-product similarity, RSA often implements correlation/cosine-based similarity. The primary consequence of this difference manifests itself in RSA's sensitivity to rotations of the data (Mehrer et al., 2020), which could be important in some scenarios, such as differing head positions in M/EEG acquisition. Kornblith et al. (2019) showed the inability of common alignment metrics to detect similar representations among identical ANN layers trained in identical scenarios but over different random seeds. This highlighted an issue that has important implications in studying model-brain alignment. The mammalian visual processing hierarchy, like ANNs, takes in representations from previous processing stages and applies computations on those representations which are further passed to downstream layers. This raised the question as to whether earlier alignment metrics might have also been insensitive to true similarities among different visual processing systems. CCA, for example, is invariant to invertible linear transformations and might not detect truly similar representations, which CKA would be able to detect. The increased sensitivity led to the consideration of CKA as a promising metric to study alignment in biological and artificial neural networks.

Han et al. (2023) use simulation experiments to test the ability of linear CKA to uncover correct ground truth models of (simulated) neural data and report that under *unreasonably* idealized conditions, CKA allows for identification capability of a reasonable degree. Recent work has shown promise in inducing brain-like representations in neural networks by incorporating various alignment metrics in custom loss functions in multi-task learning scenarios. Federer et al. (2020) minimize Representational Dissimilarity Matrices (RDMs) between model inputs and brain responses to induce the brain's statistical properties into CNNs. Pirlot et al. (2022) use Deep Canonical Correlation Analysis (DCCA) to induce brain-like responses in networks and find that they are more accurate and robust. Dapello et al. (2023) used CKA in a loss function to align neural network representations with Macaque inferior temporal cortex and found models that were better aligned to human behaviour and were more robust to adversarial examples. The above approaches that use CKA to quantify / induce alignment between biological and artificial neural networks do not specify that the debiased version of CKA was used in those experiments.

## 3 Methods

### 3.1 Brain Representations

In order to test the effect of bias correction on CKA used as an alignment metric to quantify similarity of neural network responses with human neural data, we used images, fMRI and MEG responses from the THINGS dataset (Hebart et al., 2023). This is a public repository of neural responses to a suite of naturalistic object images on relatively clear backgrounds, designed to represent solely the specified class label with as little confounding background information as possible. The fMRI data consist of three participants, while the MEG data consist of four participants. We used data from all three fMRI participants and selected participants one, two and four from the MEG data (randomly chosen). Participants in the fMRI experiment saw a set of 720 image classes (12 images in each class) from a larger image dataset consisting of 1,854 image classes. MEG participants viewed all images in the larger image dataset. The fMRI data were preprocessed using a variant of GLMSingle (Prince et al., 2022), resulting in voxelwise beta regressors per image and required no further preprocessing. Further details relating to the acquisition protocol are available in Hebart et al. (2023). Details relating to preprocessing steps we applied to the raw MEG data are given in Appendix A.2. We extracted fMRI responses across the following ROIs: V1, V2, V3, hV4, VO1, TO1, lFFA, rFFA, lPPA, rPPA, lEBA and rEBA (functional ROIs prefixed with l-/r- designate left/right brain hemispheres, while retinotopic visual regions implicitly encompass both). For the MEG data, we restrict our analysis to occipital electrodes ($N = 39$). Our choice of sensors / ROIs is motivated by capturing neural responses along the visual processing hierarchy. Further information relating to per-participant neural data dimensions is given in Appendix A.4.

## 3.2 MODEL REPRESENTATIONS

We employed two convolutional neural network (CNN) models in order to extract layer-wise activations to a subset of the THINGS stimuli set mentioned in Section 3.1. Specifically, we used ResNet18 (He et al., 2015) and CORnet-S (Kubilius et al., 2019). ResNet models have been widely used to study alignment between ANN representations and neural data (Bennett & Baldassano, 2023; Han et al., 2023). We specifically selected *ResNet18* due to its computational tractability, prominence in prior work and for its similarity in layer count to the CORnet model family, which was specifically designed to mimic properties of the mammalian visual processing hierarchy during *Core Object Recognition* and has also been widely used in brain-model alignment research (Federer et al., 2020; Han et al., 2023). We selected the *CORnet-S* variant because it had the highest score on the Brain-Score benchmark (Schrimpf et al., 2018; 2020). CORnet-S is subdivided into block-like structures: V1, V2, V4 & IT (analogous to the notion of *stages* in ResNet models). Both models were trained using the ImageNet dataset (Deng et al., 2009). Layer activations were extracted and saved using the Python toolbox *THINGSvision* (Muttenthaler & Hebart, 2021) for a single image from each of the 720 classes used in the fMRI experiment and a single image from each of the 1,854 classes used in the MEG experiment.

## 4 CKA SENSITIVITY TO RANDOM DATA

As mentioned in Section 2.2, one undesirable property is the fact standard CKA is sensitive to large discrepancies between the dimensionality of the feature spaces being compared. We use a reference matrix of random values $A \in \mathbb{R}^{1024 \times 1024}$ and compute the similarity with matrices $B \in \mathbb{R}^{1024 \times P}$ where $P$ runs from [10, 250k] on a log scale (geometric progression). We calculate similarities using: (i) biased CKA (red), debiased CKA (green) and RV2 (black). As the number of features increases, we move from the high-data low-dimensionality regime (Figure 1A) up to a commonly-occurring scenario in neural / biological data analysis where feature dimensionality vastly outnumbers the samples (Figure 1C).

As shown in Figure 1, the RV2 and debiased CKA metrics are not sensitive to large feature-sample ratio differences and accurately detect that there is no similarity among the matrix representations, since the dependence on the sample-feature ratio is removed. The debiasing step in CKA is more widely applicable to any choice of kernel, while RV2 is strictly limited to the linear case, so we do not perform any further comparisons using the RV2 metric.

Neural measurements can exhibit low signal-to-noise ratios resulting from, for example, loose sensors, misaligned voxel locations due to excessive head movement or just background activity not connected to any experimental stimuli, which has the potential to resemble random data. This fact suggests that biased CKA is not an ideal metric to directly compare among representational spaces if there is a large difference between their cardinalities. There are other instances in which inflated similarity results can arise when using biased CKA: (i) when fixing neural data (e.g. ROIs in fMRI or sensor groups / temporal windows in M/EEG) and comparing the alignment with each layer in an ANN and (ii) when fixing an ANN layer and comparing alignment across ROIs, temporal windows or sensor groups. In these scenarios the ratio of the dimensionality of the data is not fixed, which could cause issues when using biased CKA to quantify representational alignment. We examine this in the next section.

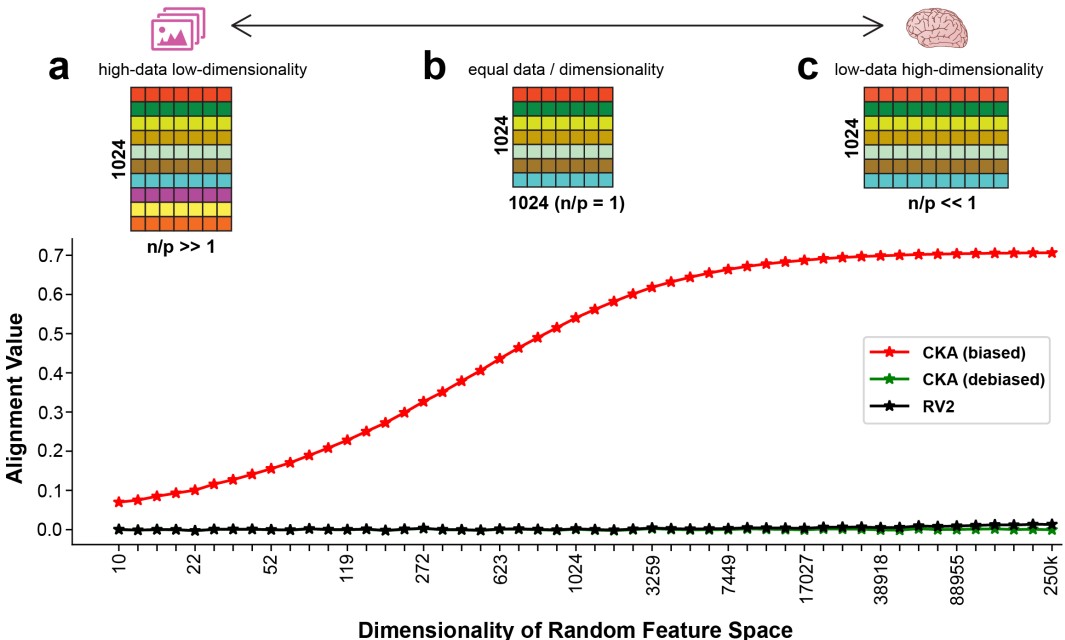

Figure 1: For a given reference matrix $A \in \mathbb{R}^{1024 \times 1024}$ sampled from a standard Normal distribution, biased CKA, debiased CKA and RV2 metrics were calculated for matrices of size $1024 \times P$, where $P$ spans a range from [10, 250k] following a geometric progression covering three domains: (a) high-data low-dimensionality, (b) equal data / dimensionality and (c) low-data high-dimensionality (most typical of neuroimaging datasets). CKA without the debiasing step is highly-sensitive to the dimensionality ratio between samples and features.

## 5  CKA SENSITIVITY TO STIMULI-DRIVEN NEURAL RESPONSES

The previous section showed how biased CKA is sensitive to changes in the ratio of sample-feature dimensionality for random matrices. We now show how this issue can lead to erroneous conclusions in a standard alignment experiment between biological and artificial neural networks across two types of brain recordings, matched on the same stimuli set and two CNNs trained on the same data (see Methods for full description). We selected primary visual cortex (V1) and occipital sensors for each of the three participants in the fMRI / MEG data, respectively. ANN responses were calculated across layers of ResNet18 and CORnet-S while keeping neural data fixed (Figure 2a-b). This was also done when fixing a CNN layer and calculating similarity scores across varying neural data sizes (fMRI ROIs in Figure 2e and increasing temporal windows of occipital MEG responses in Figure 2f). In each plot, we contrast biased CKA with debiased CKA across three data manipulations: (i) random data (See Appendix A.1), (ii) shuffling the rows of the neural data such that there is no correspondence between CNN representations and image-level neural responses and (iii) with the original data order, where sample-feature pairs are matched.

We first see that CNN representations across both ResNet18 and CORnet-S result in high alignment values to random ("neural") matrices, where the CNN representations are unchanged. This inflated result could lead to the erroneous conclusion with noisy / near-random neural data that both CNN models extract interesting representations that align well with the associated neural responses. When a CNN layer is fixed (Figure 2e-f) and we sample random data to match the size of ROIs or sensor groups, we show how conclusions based on biased CKA results alone could be misinterpreted as meaningful alignment when this effect is driven by discrepancies in feature dimensions.

For true neural responses (both in the original order and shuffled) we see increasing biased CKA alignment over layers of both CNNs. For fMRI, we see a slightly elevated value over the corresponding shuffled version, but for MEG data these values are equal. For debiased CKA, it correctly

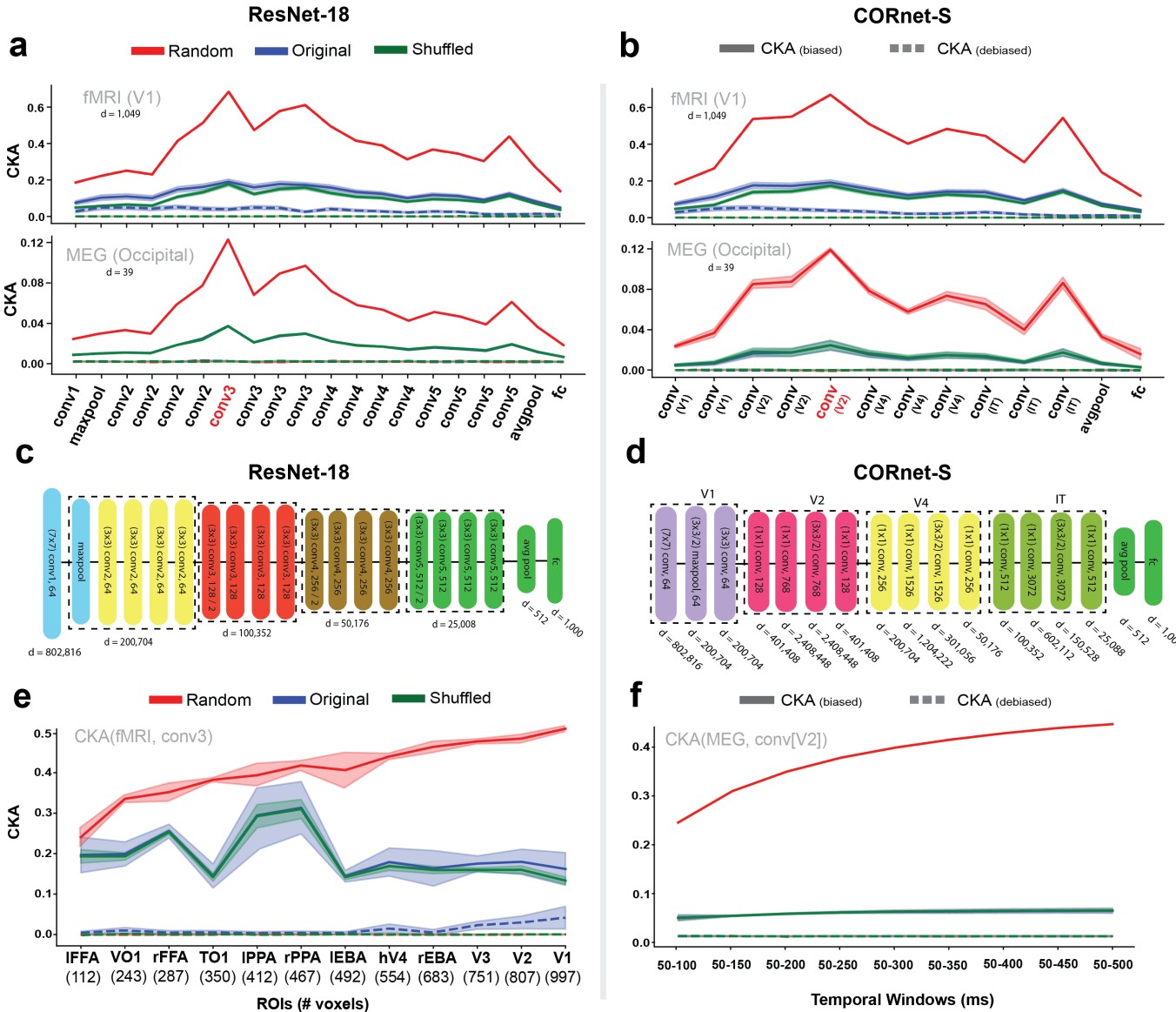

Figure 2: A-B Comparison of fMRI: V1, MEG: occipital sensors averaged over 100-150 ms window, over layers of ResNet18 & CORnet-S, in three different conditions: random data (red), original data (blue) and shuffled data (green). Results across random and shuffled conditions are the average of the three participants and 5 random seeds, while for the original data order, the average is over the three participants. C-D illustrate the layer structure of ResNet18 and CORnet-S, respectively. E-F demonstrate fixing a CNN layer and comparing representations across fMRI voxels (E) or incorporating all sensor values from increasing temporal window sizes in MEG (F).

reports zero alignment with random matrices and shuffled data. For fMRI, we do see small positive alignment values for debiased CKA in the original data order, while this is absent from the MEG data. Biased CKA without a shuffled-data control would result in a potential conclusion that both ResNet18 and CORnet-S extract similar information, however this would not be due to stimuli-driven responses. Only in the case of debiased CKA do we see a positive alignment value in this case (for fMRI). We now examine the effects of biased and debiased CKA on shuffled and unshuffled data.

## 6 CKA Sensitivity to Shuffled Data

In Figure 2 we observed that biased CKA consistently resulted in similar alignment scores for both shuffled / unshuffled data orderings across both fMRI / MEG and two CNN models (ResNet18 / CORnet-S), while debiased CKA only reported positive alignment scores for unshuffled data, i.e. the true stimuli-driven responses in the fMRI results. In this section we examine this finding in more detail and suggest an explanation for the observed results.

The *Manifold Hypothesis* states that real-world data are expected to concentrate in the vicinity of a manifold of lower dimensionality, embedded in a higher-dimensional space (Bengio et al., 2013). Deep learning models aim to learn a parameterization of the manifold of the data they are trained on (Lei et al., 2020). If a set of input data is presented to experimental participants, whose brain responses are recorded, it follows that the neural data will also lie on a lower-dimensional manifold within a higher dimensional space. In the context of alignment between biological and artificial neural networks, this can be seen when permuting neural responses and measuring the alignment with layer activations from deep learning models using a selected metric of interest. When both systems have seen the same input data, it remains an open question how much of the detectable alignment is sensitive to true stimuli-driven neural responses, as well as to a more generalized response that detects the presence of structured neural data, e.g. prototypical fMRI responses during image viewing.

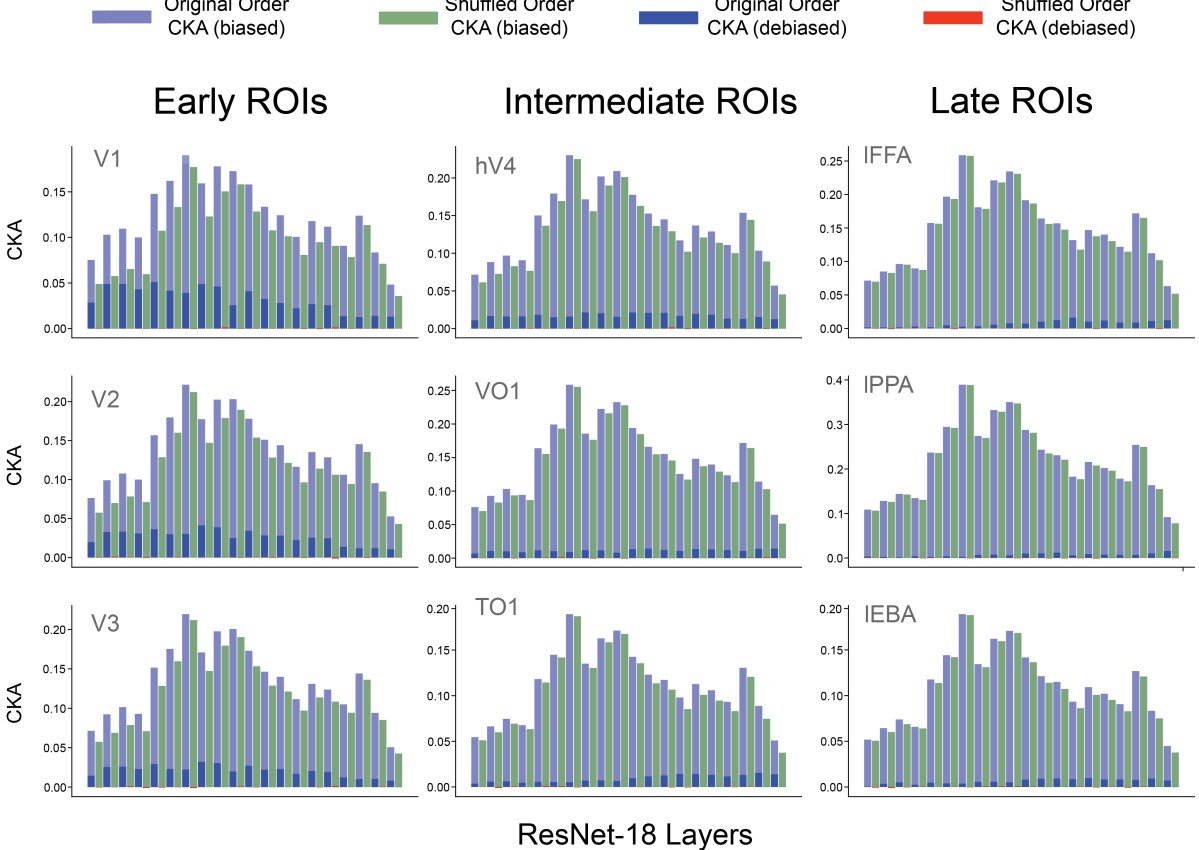

Figure 3: Across early (V1, V2, V3), intermediate (hV4, VO1, TO1) and late (left-FFA, left-PPA, left-EBA) ROIs, biased CKA is contrasted with debiased CKA across the original data order and the mean of five shuffled versions across three fMRI participants over ResNet18. Debiased CKA is the only metric that shows sensitivity to stimuli-driven responses. This replicates the pattern reported in earlier work that lower visual areas exhibit greater correspondence to lower-layer CNN representations and similarly for higher visual areas / higher-layer CNNs (Yamins et al., 2014).

Figure 3 shows a comparison between biased and debiased CKA across ResNet18 (CORnet-S results reported in Appendix A.6) for a subset of fMRI ROIs, categorized into early (V1, V2, V3), intermediate (hV4, VO1, TO1) and late (lFFA, lPPA, lEBA) groups (Witthoft et al., 2013). For the original data order, the reported value is the mean across the three fMRI participants ($N = 3$), while for the shuffled order, the reported value is the mean across five random shuffles per participant ($N = 3 \times 5 = 15$). For all ROIs and ResNet18 layers, biased CKA indicates an alignment for the original data order which is only slightly larger than for shuffled data. This indicates that biased CKA is not primarily sensitive to stimuli-driven responses. Debiased CKA does not suffer from this undesirable side-effect and returns a positive value only for stimuli-driven neural responses. It is only via debiased CKA that we recover previously-reported observations that early visual regions are more aligned with lower layers of CNNs, while later regions are more aligned with the upper layers (Yamins et al., 2014; Yamins & DiCarlo, 2016). Though this pattern is clear, the values we report are small in magnitude compared to results obtained with biased CKA, which we expect might be associated with the limited dataset of 720 images. We performed an additional analysis (See Appendix A.7) between fMRI areas V1 & left PPA (lPPA) and the first three layers of ResNet18. We gradually varied the amount of shuffling and see that only in the case of positive stimuli-driven alignment scores is there sensitivity to the amount of shuffling (in V1). For lPPA, biased and debiased CKA are insensitive to varying amounts of shuffling. This supports our interpretation that biased CKA is not sensitive to stimuli-driven activity.

An interesting observation in Figure 3 is that with respect to the original data order, biased CKA alignment exhibits an increased value over the corresponding shuffled version in line with the magnitude of the debiased CKA score. This is predominantly seen in the early ROIs (V1, V2 & V3), while when debiased CKA does not detect alignment (e.g. lPPA), biased CKA is near-identical for both the original and shuffled data orders. This indicates that when stimuli-driven alignment exists between biological and ANNs, biased CKA is able to detect it as roughly the additional alignment given over and above that which is given for shuffled data. We have posited a hypothesis to explain what biased CKA might be additionally sensitive to, in light of the similarity to shuffled sample-feature pairs. As data exhibit more of an unstructured random profile, biased CKA scores rapidly increase (See Figure 1). The similarity between shuffled and unshuffled data pairs points to a sensitivity based on the structure of the neural data itself, which might be characterized by structure that *looks like fMRI data*, for example. Even when using random data sampled from Normal distributions where voxel-wise / sensor-wise means and standard deviations are used to parameterize these distributions (See Appendix A.1) in order to closer approximate structured neural data, biased CKA results are indistinguishable from random results reported in Figure 2. The fact that shuffled responses are treated differently lends further support to our idea that biased CKA sensitivity is driven by a generalized neural response pattern.

## 7 CONCLUSION

Using fMRI and MEG data paired with stimuli from the THINGS dataset, we showed that measuring alignment with biased CKA has several weaknesses that are corrected by implementing a debiasing step. Biased CKA can be increased arbitrarily by varying the feature ratio w.r.t. a fixed dataset size (e.g. examining fMRI ROIs w.r.t. to a single CNN layer). This prevents direct comparisons across varying feature dimensionalities which often arises in comparisons between neural data and ANNs (e.g. different ROI sizes / sensor groups, temporal windows, parameter counts). We postulate that biased CKA is sensitive in part to the inherent structure of neural data since biased CKA reveals similar alignment values between shuffled and unshuffled data when debiased CKA results are close to zero. We show that alignment based on the true stimuli-driven responses is only obtained when using the debiased version of CKA. Our analysis did not show debiased CKA alignment with MEG data. We chose one method of preprocessing, but other methods that boost the signal-to-noise ratio (e.g. more data, different settings) will likely increase our ability to detect stimuli-driven alignment with ANN representations. It will be informative to quantify the same analysis across other common alignment metrics, which we leave to future work. As biological and ANN alignment research is an emerging topic in representation learning, methods like CKA are being used not only to quantify alignment but also to induce brain-like representations in neural networks. We have demonstrated some issues when using CKA as an alignment metric in the hope that the community can sidestep erroneous conclusions and spot potential issues when assessing reported alignment metrics in their work and the work of others.

ACKNOWLEDGMENTS

We would like to thank Jessica Thompson for a constructive discussion that has better informed our work.

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

# A APPENDIX

## A.1 GENERATING RANDOM DATA

In order to quantify the effect of CKA in its biased and debiased form on random data for a specific fMRI ROI or MEG time window / sensor group, we sampled from a multivariate standard Normal distribution of the same shape, where each entry is independent of the other. Our later analyses suggested structure of the neural data is in part responsible for biased CKA scores (as seen with shuffled data). We then repeated our random analyses where we estimated each voxel / sensor dimension by taking the mean over responses to the entire image set (and similarly for the standard deviation). We then repeated the random analyses by sampling from Normal distributions that were more similar to the structure of the neural data and confirmed that our results did not differ from sampling from the original case when using Standard Normal distribution.

## A.2 MEG PREPROCESSING & SENSOR SELECTION

We used the raw version of MEG data in the THINGS database and wrote custom code to apply preprocessing on each session. Specifically, we band-passed filtered the data between [0.1, 40] Hz using a firwin filter and extracted a window of [- 0.1, 1.5] second around each stimulus event, baseline-corrected each epoch using the 100 ms pre-stimulus interval by subtracting the mean and downsampled the data to 200 Hz. We added a mapping to a predefined train, validation and test split designed in a separate analysis and concatenated MEG responses associated with the data set reported in Section 3.1. We did not perform analyses to detect bad segments or channels, with the exception of dropping MRO11 from all participants as Hebart et al. (2023) advised this due to issues observed during the original data acquisition session that resulted in this channel being excessively noisy. We extracted all MEG channels that were labelled as O[ccipital] in the original data structure for our analyses, resulting a feature dimension of $N = 39$ for each time step of MEG data.

### A.3 CODE IMPLEMENTATION

Our code was written in Python / PyTorch and we used the implementation given in Kornblith et al. (2019) to calculate biased & debiased CKA. The code used to run the analysis and generate the figures in this work are available at the following link: `https://github.com/Alxmrphi/correcting_CKA_alignment`.

### A.4 PER-PARTICIPANT FUNCTIONAL MRI ROI SIZES

|  | **V1** | **V2** | **V3** | **hV4** | **VO1** | **TO1** | **lFFA** | **rFFA** | **lPPA** | **rPPA** | **lEBA** | **rEBA** |
|---|---|---|---|---|---|---|---|---|---|---|---|---|
| **Sub 01** | 1049 | 774 | 762 | 613 | 287 | 369 | 154 | 399 | 395 | 414 | 563 | 640 |
| **Sub 02** | 1104 | 988 | 830 | 577 | 266 | 364 | 124 | 276 | 614 | 582 | 180 | 531 |
| **Sub 03** | 839 | 660 | 663 | 472 | 218 | 318 | 60 | 187 | 228 | 405 | 735 | 879 |
| **Average** | 997 | 807 | 751 | 554 | 243 | 350 | 112 | 287 | 412 | 467 | 492 | 683 |

### A.5 UNBIASED HSIC KERNEL-CENTERING FORMULA

Székely & Rizzo (2014) introduced the following formula to compute the unbiased HSIC estimator for an $n \times n$ matrix, $A$. The only requirements are: (i) the main diagonal contains 0s and (ii) $n > 2$. The unbiased centered matrix, $\tilde{A}$ is defined below.

$$\tilde{A}_{i,j} = \begin{cases} a_{i,j} - \frac{1}{n-2} \sum_{\ell=1}^{n} a_{i,\ell} - \frac{1}{n-2} \sum_{k=1}^{n} a_{k,j} + \frac{1}{(n-1)(n-2)} \sum_{k,l=1}^{n} a_{k,l} & \text{if } i \neq j \\ 0 & \text{if } i = j \end{cases} \quad (4)$$

## A.6   Results for CKA Sensitivity to Shuffled Data (CORnet-S)

Figure 4: Across early (V1, V2, V3), intermediate (hV4, VO1, TO1) and late (left-FFA, left-PPA, left-EBA) ROIs, biased CKA is contrasted with debiased CKA across the original data order and the mean of five shuffled versions across three fMRI participants over CORnet-S. Debiased CKA is the only metric that shows sensitivity to stimuli-driven responses and with it we replicate the previously-reported pattern that lower visual areas exhibit greater correspondence to lower-layer CNN representations and similarly for higher visual areas / higher-layer CNNs.

### A.7 Results for CKA Sensitivity to Partially Shuffled Data (ResNet18)

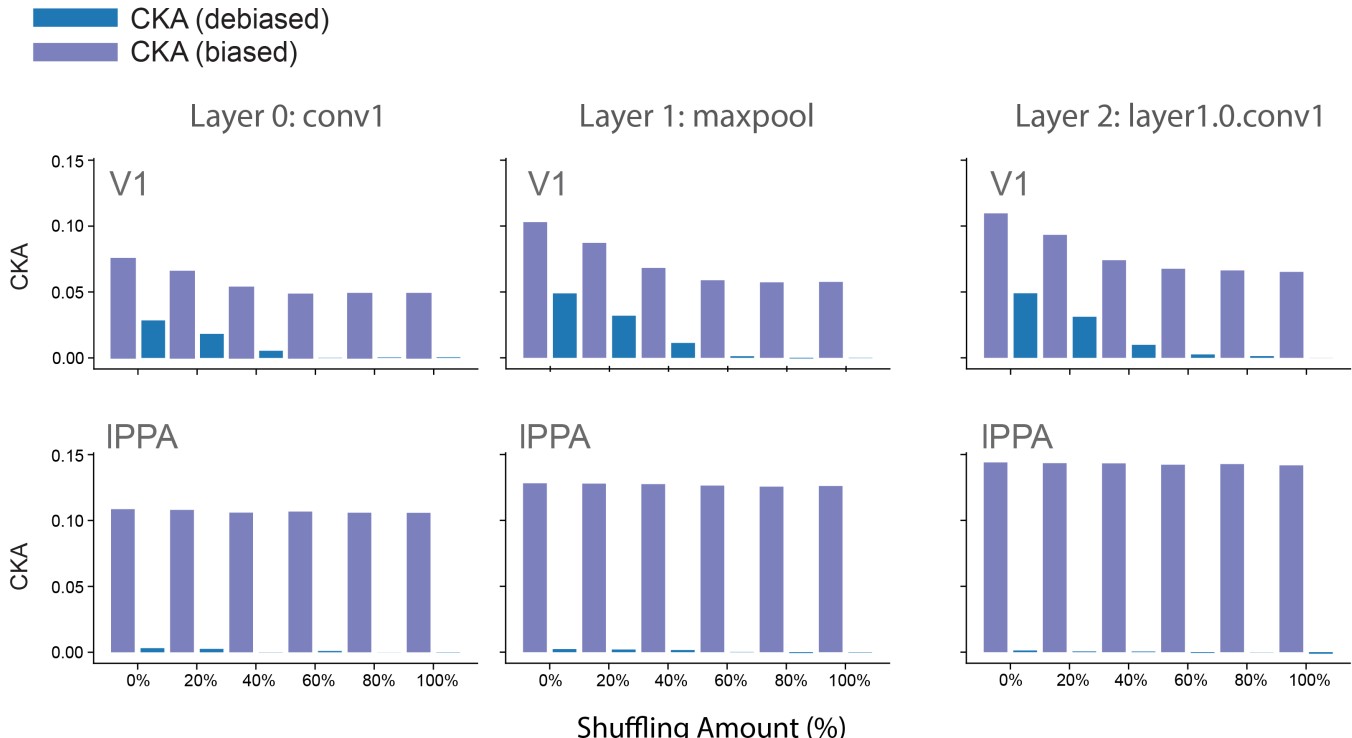

Figure 5: When no stimuli-driven alignment is detected (via debiased CKA) between brains and biological neural networks, biased CKA aligns equally as well to shuffled (neural) data as it to does to the original ordering. In ROIs where this occurs, no change is expected as the amount of shuffling is varied from 0% to 100%. Conversely, areas that do appear to contain stimuli-driven alignment should gradually decrease in alignment score as the amount of shuffling is increased. From Figure 4 we selected V1 and left PPA (lPPA) to be representative of the two cases (V1: sensitive to stimuli-driven alignment, lPPA: insensitive to stimuli-driven alignment). We incrementally varied the percentage of shuffled items and measured the alignment in the first three layers of ResNet18 (conv1, maxpool & layer1.0.conv1). We find that the amount of shuffling has no effect on ANN-ROI pairs that originally showed no stimuli-driven alignment. We hypothesize that this is due to biased CKA's sensitivity to a generalized neural response. For early layers of ResNet18 and V1 in Figure 3, we saw stimuli-driven alignment scores, which are indeed gradually reduced as the percentage of neural data shuffling is increased.

