# OpenReview forum: "Correcting Biased Centered Kernel Alignment Measures in Biological and Artificial Neural Networks"
_ICLR.cc/2024/Workshop/Re-Align — ICLR 2024 Workshop Re-Align Poster_

### Official Review · Reviewer_VBcw · 2024-02-20
**A potentially highly relevant paper.**

**Rating:** 3
**Fit:** 3
**Confidence:** 2

**Workshop Review:**

## Summary
This paper demonstrates the importance of using debiased CKA instead of standard CKA in the context of representational alignment between neural data and ANN activations. The authors point out that neural data is typically in the “low-data, high-dimensionality”-domain, in which standard CKA is systematically biased, thus overestimating alignment. They demonstrate this effect empirically with random matrices of varying feature-sample ratios, showing that CKA converges to a high alignment value as the ratio grows, while debiased CKA correctly maintains its low value. Next, the authors examine CKA values between real fMRI / MEG measurements and a ResNet18 / CORnet-S, showing how CKA drastically overestimates the alignment even for random and shuffled “neural” values, while debiased CKA only takes non-zero values for the real data.

## Strengths
The paper makes a potentially highly relevant contribution to the field of representational alignment, demonstrating the practical relevance and utility of debiased CKA. Furthermore, the paper is clearly written, well-structured and technically sound, as far as I can tell. I appreciate the informative overview of CKA instead of a typical related work section. While the paper does not present a new method or collect new data, it does open a new perspective on existing data and raises a relevant point.

## Weaknesses
Section 5 is very descriptive of figure 2, but I’m actually very intrigued by the implications of findings in figure 2. If the value range of debiased CKA is [0,1], then these results effectively imply that there is very little alignment between ANN activations and fMRI data, and almost no alignment between ANN activations and MEG data - even though previously, one might have believed this to be the case, based on the high standard CKA values. This goes far beyond just noting that debiased CKA is less sensitive to the failure modes of shuffled and random data. Essentially, a bit more context for these results would be nice.

## Recommendation
As somebody not very familiar with CKA, I don’t know whether the community has effectively already made the shift to debiased CKA, or if the practice of using standard CKA is still common. Especially in the latter case, the paper constitutes an important piece of information for practitioners, possibly shedding new light on earlier results. I propose to accept.

## Questions
- What is the value range of (debiased) CKA? Does it yield values in [0,1]? Explaining this would be very helpful to put the findings in figures 3 and 4 into perspective.
- Are there other works which report high standard CKA values and therefore draw conclusions that may now turn out to be erroneous? I am wondering about the practical impact of this finding (e.g. if nobody uses standard CKA, for this reason or others, this finding seems less relevant than if multiple other findings are now called into question).
- On page 8 you write that the small absolute values obtained with debiased CKA could be explained by the small size of the dataset, but as far as I understand, the biased CKA values were obtained using the same set of 720 images, right? So why should the number of images explain this effect, vs. it being a “true” effect, that maybe brains and ANNs are not as aligned as biased CKA leads us to believe? Maybe it would be interesting to quantify how many images one needs to get reliable debiased CKA values, or get some other measure of uncertainty.

## Additional Feedback
- It is a bit hard to distinguish lines in figure 2, it took me a while to realize that debiased CKA for random data was not actually omitted, but just sits at 0. Maybe plotting with less alpha would help, or using other linestyles. Also, the color scheme is not ideal for colorblind folks.
- What would the debiased CKA for fMRI-data of different participants be? The idea being to establish a noise ceiling - if brain-to-brain alignment is also only 0.2, this puts the absolute debiased CKA values into perspective and says something about the signal-to-noise ratio of fMRI in general.
- “Previous research has noted that this situation is where standard (”biased”) CKA becomes an unreliable alignment metric that can potentially lead to erroneous conclusions.” $\rightarrow$ maybe cite that other work here
- “Kornblith et al. (2019) showed that neural networks trained using the same architecture and hyperparameters, but across different random seeds, were unable to detect similarity between corresponding layers in several common similarity metrics, but not CKA.” $\rightarrow$ this sentence is muddled, do you mean: “Kornblith et al. (2019) showed that several common similarity metrics were unable to detect similarity between corresponding layers of neural networks trained using the same architecture and hyperparameters, but different random seeds. In contrast, CKA was able to detect this similarity.”?
- “The above approaches that use CKA to quantify / induce alignment between biological and artificial neural networks specify a form that suggests the standard (biased) implementation was used in those works.” $\rightarrow$ “specify a form” sounds a bit unclear here, are you just saying that they probably used the biased implementation, but you’re not sure? It seems pretty relevant to me whether people already know about this issue and are already using debiased CKA or not. You could just ask them what they did.
- “It will also be informative to quantify the same analysis across a host of other common alignment metrics and, which we leave to future work.” $\rightarrow$ remove “and”
- “ANN responses were calculated across layers of ResNet18 and CORnet-S while keeping neural data fixed (Figure 2a-b) and also when fixing a CNN layer and calculating similarity scores across varying neural data sizes (fMRI ROIs in Figure 2e and increasing temporal windows of occipital MEG responses in Figure 2f).” $\rightarrow$ maybe break this into two sentences to increase clarity.
- “CCA, for example, is invariant to invertible linear transformations and might not detect truly similar representations, which CKA would be able to detect.” $\rightarrow$ isn’t CCA’s invariance a desirable property here? If CCA is invariant, that means it would detect that representations are truly similar.
- On page 7, you refer to figure 4 but mean figure 3, probably the labels of the two figures (ROIs for Resnet18 / CorNet-S) got mixed up?

**Reason For Not Giving Higher Score:**

N/A

**Reason For Not Giving Lower Score:**

The paper demonstrates a serious issue with standard CKA. Using debiased CKA instead seems to be something that other researchers should know about. While the method itself is not new, it appears to me that this convincing demonstration of the need to actually use the debiased version could shed new light on other results.

**Reviewer Domain:**

cognitive science

---

### Official Review · Reviewer_ZQXK · 2024-02-23
**Review for "Correcting Biased Centered Kernel Alignment Measures in Biological and Artificial Neural Networks"**

**Rating:** 2
**Fit:** 3
**Confidence:** 2

**Workshop Review:**

Summary: This paper presents a methodical evaluation of CKA for alignment of representations between biological (fMRI and MEG) and artificial neural networks. The evaluation of biased and debiased CKA was presented in a well-motivated fashion and well-situated in the representational alignment literature. The authors' motivation for using debiased CKA over the ordinary [biased] CKA was well-supported by their analyses. Overall, I found this paper to be interesting and clearly addressed the points the authors set forth; I have some recommendations for improvement that I outline below.

Strengths:
- A well-suited and motivated dataset choice (THINGS fMRI and MEG) for the problem the authors sought to address
- Reasonable ANN choices and demonstrated generalization across ANNs
- Clear motivation for why biased CKA needs addressing
- Shuffled data and random data were adequate control comparisons
- Clearly written and comprehensive paper

Areas for improvement:
- It was not entirely obvious how the debiased CKA is formulated. Since it is the backbone of the paper, I think the authors should include a formulation of the metric in section 2.2.
- The point about linear kernels was a bit confusing. In the final paragraph of section 2.2, the authors note that the debiasing step added to CKA is not restricted to linear kernels, which makes debiased CKA more useful than the RV2 coefficient, but in section 2.1 the authors note that "it is virtually ubiquitous to use CKA with a linear kernel." Does debiased CKA with a linear kernel afford anything beyond RV2? If they are not identical, I would want to see a comparison with RV2 at least in Figure 2.
- Figures 2 (A,B,E,F), 3, and 4 are very difficult to read given the colors and overlays. If the authors think that the best way to visualize the data are to have them as overlaid line/bar plots with common axes, and I understand this given the point the results are trying to drive home, I would highly recommend reconsidering the colors / saturations used for each comparison  (original CKA/debiased CKA and shuffled CKA/debiased CKA). As it currently is, it's impossible to see the CKA debiased results.
- RSA would be an easy additional benchmark to throw into these analyses, and I think the first necessary additional comparison.
- A more detailed addressing of the MEG results would be nice. MEG preprocessing pipelines and signal processing is far less developed than fMRI, so it is unshocking to me that the results are less conclusive. However, a more thorough discussion of this point would be valuable, perhaps proposing what specific preprocessing could be changed, why those choices would be relevant, how the sensors were chosen for the current analysis. Could it be that MEG just does not show positive debiased CKA alignment with ANNs, regardless of the preprocessing? These would be valuable discussion points.

**Reason For Not Giving Higher Score:**

The topic matter is clearly relevant for the workshop and important to discuss, but my impression is that it is not substantially novel/impactful to warrant a talk.

**Reason For Not Giving Lower Score:**

The work is well-situated in the literature and addresses an important question in how we evaluate the alignment of representations.

**Reviewer Domain:**

neuroscience

---

### Official Review · Reviewer_Rhi5 · 2024-02-23
**Nice work**

**Rating:** 3
**Fit:** 3
**Confidence:** 2

**Workshop Review:**

The authors in this work argue for debiasing the centered kernel alignment (CKA) measure. Specifically, the authors consider a setting where researchers use CKA to compare the representational similarity between biological and artificial neural networks. The authors use convincing results on simulated, MEG, and fMRI data to make their point, and show the reader that results without debiasing CKA can completely obfuscate the actual representational similarity between biological and artificial neural networks.

**Clarity** \
The authors do a great job of convincing the reader how important it is to use debiased CKA through a variety of experiments. To improve the paper for a potential full conference or journal submission, I would recommend a few things to improve the clarity:
- On Page 4, the sentence before the Methods section reads: “The above approaches that use CKA to quantify/induce alignment between biological and artificial networks specify a form that suggest the standard (biased) implementation was used in those works.”. Although I generally agree it is good to point out potential biases in other people’s works, I do think this statement is a little too implied for the actual experiments you have performed. I understand you reference these works to explain why it is important to bring awareness to debiased CKA, since these works likely still (accidentally) use a biased version of CKA. This can certainly be true, but I believe it would be more constructive in a full conference or journal paper to reproduce their experiments with debiased CKA to verify whether their claims hold up, and how the new results differ. Otherwise I would reword/remove this sentence because it can be read as questioning their conclusions, which the experiments in this paper do not directly tackle.
- On Page 1 “Previous research has noted that this situation is where standard (“biased”) CKA becomes an unreliable alignment metric that can potentially lead to erroneous conclusions.”. It would be helpful to cite this research.
- Figure 2, I think it would help the reader if the y-axes for a and b are log-scale, and the y-axes are aligned between the ResNet-18 and CORnet-S. Currently, for the MEG results, the axes differ between a and b, which makes it harder to compare the results between the two models.

**Correctness** \
All the analyses support the main conclusion in the paper, and seem to be conducted in a clear and reproducible manner. Especially if the authors manage to publish the code upon acceptance, which they promise to do in the appendix.

**Novelty** \
Although the importance and idea of debiasing CKA were introduced in Kornblith et al. 2019, which the authors themselves cite, the analyses in that paper mostly compare representations between layers in the same network. The authors in this work specifically evaluate representational similarity between biological and artificial neural networks. Especially to draw more attention towards explicitly debiased CKA for biological - artificial neural network representational similariy, this work is important. To improve the paper for a potential full conference or journal submission, I would recommend a few things to improve the novelty:
- Page 9, Apart from the first point I mentioned under clarity, I believe the authors can unpack their last sentence before the conclusion more in a full conference or journal paper: “The fact that shuffled responses are treated differently lends further support to our idea that biased CKA sensitivity is driven by a generalized neural response pattern.”. I believe showing this more rigorously through experiments will add to the novelty of the paper. Additionally, if the authors decide to pursue a direction like this, I would recommend moving Figure 3 to the Appendix to reclaim some space. Although it is an interesting result, especially for a workshop paper, I believe it is too similar to Figure 2 to add much to the novelty of the work in a full conference or journal paper.

**Interest to the community** \
I think this work is a perfect fit for the workshop.

**Grammatical/spelling errors (these may be personal preference so feel free to ignore them)** \
- P4, “… found models that were better aligned to human behaviour were more robust” -> human behaviour and were
- P5, “Both models were both trained” -> Both models were trained
- P5, “As the number of features increases, we move from high-data low-dimensionality regime…” -> from the high-data low-dimensionality regime
- P5 “…where feature dimensionality vastly outnumbers the number of samples…” -> vastly outnumbers the samples

**Reason For Not Giving Higher Score:**

I gave the authors the highest score for the workshop.

**Reason For Not Giving Lower Score:**

Although I believe there are some limitations to the paper, including a slight lack of novelty, I believe the importance of presenting this work at a workshop on representational alignment outweighs the negatives. Furthermore, many of my critiques are under the assumption that the authors want feedback on their work for a future archival publication :-)

**Reviewer Domain:**

machine learning

---

### Decision · Program_Chairs · 2024-03-02

Accept (Poster)